# OpenReview forum: "Graph-Enhanced Learning for Predicting Optimal Drug Combinations Using Contrastive Embedding"
_ICLR.cc/2025/Conference — Submitted to ICLR 2025_

### Official Review · Reviewer_kSVB · 2024-10-27

**Soundness:** 2
**Presentation:** 2
**Contribution:** 2
**Rating:** 3
**Confidence:** 4

**Summary:**

This work presents a new framework for the problem of drug-drug interaction prediction. In the framework, this work integrates the theory of domain adaptation into the DDI problem, provides proofs for the properties of the framework. In experiments, the proposed framework outperforms other GNN methods for the problem of DDI prediction.

**Strengths:**

1. This work introduces the domain adaptation theory for DDI problem, providing a new perspective to handle that problem.

**Weaknesses:**

1. Although this work focuses on theoretical property of the proposed DDI-DA bundle, a large amount of proof process is simplified. For example, “we show” and “we prove” are used instead of detailed proofs of the properties. These detailed proofs are also not provided in appendix.
2. This work focuses more on machine learning theoretical proofs of DDI-DA bundle. However, this work does not show how these theoretical proofs are connected to DDI problem and drug discovery.
3. The experiment part is too simple. The experiment results do not show how they are related to “domain adaptation”. Moreover, the characters in the figures are too small.
4. In introduction part, it is mentioned that “our framework opens new avenues for research in a wide range of scientific disciplines”, but this part is not confirmed in the following contents of this paper.

**Questions:**

1. Although there is a series of theoretical properties for proposed framework in section 2, they are not well verified in the experiment part.
2. In the related works and experiments, existing DDI methods that utilize GNN are not mentioned and compared.
3. In the introduction, it is mentioned that “Our unified theory is flexible and can accommodate a wide range of underlying model architectures”. The contribution is not well verified in the paper.
4. There is a fault for the framework of the paper: “DATASET” should be a subset of “EXPERIMENT”, but “EXPERIMENT” is section 2.1 and “DATASET” is section 2.2.

---

### Official Review · Reviewer_bD8H · 2024-11-03

**Soundness:** 2
**Presentation:** 1
**Contribution:** 3
**Rating:** 3
**Confidence:** 2

**Summary:**

The paper introduces a new formulation of domain adaptation for drug
interaction prediction. It is based on optimal transport and
information geometry, to which it incorporates existing measurements
of drug-drug interaction.

**Strengths:**

The Drug-Drug interaction prediction problem is a difficult problem
and important, a key to combination therapies. This is motivated well
in the introduction.

The introduced methods and theory in Section 2 seem rigorous,
interesting, and potentially exciting.

**Weaknesses:**

Explanation of both the theory and the application include very little
tutorial. The paper is currently only understandable to readers fully
familiar with both, which is arguably a very narrow audience.

Explanation of the actual learning and inference methods is severely
inadequate, starting from the key Figure 1 which has not been
explained at all.

The relationship between the neural learning and inference tasks and
the theory remains unclear.

The application task is unclear; the machine learning setup has not
been explained at all.

Some hyperbole in the abstract and introduction. There may be grounds
of making the claims the paper makes, but they are not explained
sufficiently well to become clear to the reader.

Quantum: not mentioned after the intro

The method has not been compared against existing DDI prediction
methods

**Questions:**

See the Weaknesses: Could you clarify any of those?

What are the "drug distributions" mentioned in the intro?

---

### Official Review · Reviewer_NRWk · 2024-11-03

**Soundness:** 2
**Presentation:** 1
**Contribution:** 2
**Rating:** 3
**Confidence:** 2

**Summary:**

The manuscript presents a theoretical framework that introduces a novel optimal transport formulation, specifically designed to preserve drug-drug interactions (DDIs). This formulation is then cast as a geodesic problem within the structure of Finsler Geometry, creating a tool for addressing DDI-aware domain adaptation. The authors apply this framework to the task of predicting drug-drug interactions, an area of significant relevance in biomedical sciences, potentially improving the robustness and transferability of synergy prediction models across diverse disease domains.

**Strengths:**

- The proposed approach systematically improves drug interaction prediction across multiple Graph Neural Network (GNN) architectures, including GIN, GraphSAGE, GAT, and GCN. This suggests the framework’s robustness and flexibility for various graph-based learning applications in biomedical settings
- By embedding DDI constraints into the optimal transport cost function, the authors propose a fresh approach that unites domain adaptation with interaction-aware learning.

**Weaknesses:**

- The manuscript does not clearly explain the input features used for each drug, making it unclear if the authors are using molecular SMILES strings converted into graphs (as Figure 1 might suggest). Without this, it’s difficult to evaluate the validity of their choices in modeling and formulating the optimal transport problem. Also, attempts to run the code were unsuccessful due to missing preprocessing files, preventing direct verification of the input pipeline.
-  Due to the lack of feature detail, it’s challenging to gauge whether the DDI-aware optimal transport formulation balances complexity with practical utility. Specific reasoning behind certain choices in this complex formulation would help in evaluating its practical value.
- The manuscript mentions quantum information theory in the introduction but doesn’t explain how it relates to the formulation or theory
- The authors use a custom benchmarking approach with random negatives or TWOSIDES, but they don’t compare or contrast it with other approaches on this topic. Adding a baseline, as outlined in recent reviews https://academic.oup.com/bib/article/24/4/bbad235/7217116, would provide more context for this approaches' performance and impact.
- The results combine the effects of domain adaptation and contrastive learning without a separate analysis of each. Isolating each component’s impact would make it easier to understand how each technique contributes to the model’s performance.

**Questions:**

- Please add a section detailing the task of drug-drug interaction prediction, specifying in detail input features and how the framework applies to this task.
- Make sure the code repository works by adding the missing preprocessing scripts so that others can replicate the data generation and model training.
- Perform an ablation study in Tables 2 and 3 that separates the effects of domain adaptation and supervised contrastive learning, allowing readers to see the individual contributions of each.
- Please Avoid/Remove subjective words like “cutting-edge” and “revolutionary,” which can detract from the rigor of the analysis. Stick to objective descriptions.
- Please Add a state-of-the-art baseline to give a clear benchmark https://academic.oup.com/bib/article/24/4/bbad235/7217116

---

### Official Review · Reviewer_7Nve · 2024-11-04

**Soundness:** 1
**Presentation:** 1
**Contribution:** 1
**Rating:** 1
**Confidence:** 3

**Summary:**

The authors formulate a DDI-aware optimal transport problem, which involves a geodesic equation on an infinite-dimensional Finsler manifold. They propose to use this formulation for drug interaction predictions.

**Strengths:**

The strengths are that the results section seems to show that their methods out-compete the others. They were also able to formulate interesting theorems that in an abstract way generalizes drug interactions.

**Weaknesses:**

The relation between the theory and their implementation is completely lacking. It is difficult to understand how to go from the theory to the implementation and thus difficult to see  if it is correct. No proofs were given sometimes! They say 'The full proof requires careful analysis of the regularity of solutions and the use of techniques
from optimal control theory in spaces of measures, as well as the theory of nonlocal interaction
equations' and then fail to give the proof.

**Questions:**

Where are the proofs?
How does the theory relate to the implementation?
How does your method improve upon other previous work?

---

### Meta-Review · Area_Chair_mf2h · 2024-12-17

**Metareview:**

All reviewers identifies several (sometimes severe) weaknesses, problems and open questions. Critical points included the lack of formal proofs, the unclear connection between theory and real-world settings and the lack of baseline experiments. There was no rebuttal, so I recommend rejection of this paper.

**Additional Comments On Reviewer Discussion:**

There are many open questions, but no rebuttal...

---

### Decision · Program_Chairs · 2025-01-22

Reject